# Approaches of Combining Machine Learning with NMR-Based Pore Structure Characterization for Reservoir Evaluation

Wenjun Zhao [1], Tangyan Liu [2,*], Jian Yang [3], Zhuo Zhang [1], Cheng Feng [2] and Jizhou Tang [1]

[1] State Key Laboratory of Marine Geology, Tongji University, Shanghai 200092, China; 17551025644@163.com (W.Z.); zzhang6@tongji.edu.cn (Z.Z.); jeremytang@tongji.edu.cn (J.T.)
[2] Faculty of Petroleum, China University of Petroleum-Beijing at Karamay, Karamay 834000, China; fcvip0808@126.com
[3] Engineering Technology Research Institute of Southwest Oil & Gas Field Company, PetroChina, Chengdu 610017, China; yj08@petrochina.com.cn
* Correspondence: tyliu05169@tongji.edu.cn

**Abstract:** Tight gas, a category of unconventional natural gas, relies on advanced intelligent monitoring methods for their extraction. Conventional logging for reservoir evaluation relies on logging data and the manual setting of evaluation criteria to classify reservoirs. However, the complexity and heterogeneity of tight reservoirs pose challenges in accurately identifying target layers by using traditional well-logging techniques. Machine learning may hold the key to solving this problem, as it enables computers to learn without being explicitly programmed and manually adding rules. Therefore, it is possible to make reservoir evaluations using machine learning methods. In this paper, the reservoir quality index (RQI) and porous geometric parameters obtained from the optimized inversion of the spherical–tubular model are adopted to evaluate the reservoir. Then, three different machine learning approaches, the random forest (RF) algorithm, support vector machine (SVM) algorithm, and extreme gradient boosting (XGB) algorithm, are utilized for reservoir classification. The selected dataset covers more than 7000 samples from five wells. The data from four wells are arranged as the training dataset, and the data of the remaining one well is designed as the testing dataset to calculate the prediction accuracies of different machine learning algorithms. Among them, accuracies of RF, SVM, and XGB are all higher than 90%, and XGB owns the highest result by reaching 97%. Machine learning based approaches can greatly assist reservoir prediction by implementing the well-logging data. The research highlights the application of reservoir classification with a higher prediction accuracy by combining machine learning algorithms with NMR-logging-based pore structure characterization, which can provide a guideline for sweet spot identification within the tight formation. This not only optimizes resource extraction but also aligns with the global shift towards clean and renewable energy sources, promoting sustainability and reducing the carbon footprint associated with conventional energy production. In summary, the fusion of machine learning and NMR-logging-based reservoir evaluation plays a crucial role in advancing both energy efficiency and the transition to cleaner energy sources.

**Keywords:** machine learning; intelligent evaluation and prediction; spherical–tubular model; petrophysical parameters

## 1. Introduction

As the global demand for sustainable energy grows rapidly, the utilization and storage of deep green energy resources (conventional and unconventional gas resources, geothermal energy, etc.) are becoming more and more significant [1–3]. Reservoirs, as one of the underground spaces, are not only a development and production area, but also regarded as a favorable place for geological energy storage and greenhouse gas sequestration [4–6]. Reservoir classification and evaluation is pivotal in understanding the geological structure and reservoir characteristics, assessing the energy reserves, and thereby ensuring

effective exploitation while reducing environmental risks [7]. The reservoir assessment in well logging is a challenging job that involves reservoirs classifying based on specific well-logging data and manually setting rules [8,9]. Traditional methods of reservoir evaluation often fail to meet the demands for accuracy and efficiency. The machine learning algorithms in this context can help explore new rules and uncover hidden patterns within the data, which may be difficult or impossible when only conventional methods are applied. Importantly, the integration of machine learning and reservoir evaluation greatly enhances reservoir characterization, identifies productive zones, and optimizes well placement and production strategies [10–13]. This synergy also aligns with the broader global shift towards sustainable energy development and the mitigation of greenhouse gas emissions associated with traditional energy production.

As the world strives for more sustainable and cleaner energy sources, the evaluation and development of tight oil and gas reservoirs in China, characterized by low porosity and low permeability, pose formidable challenges [14,15]. In order to better understand and optimize development, scholars have conducted extensive research on the evaluation of tight oil reservoirs and gas-bearing reservoirs. Yang et al. (2013) proposed a five-parameter method to evaluate low-permeability reservoirs with the main throat radius, movable fluid percentage, pseudo-threshold pressure gradient, crude oil viscosity, and clay mineral content [16]. Most researchers calculate porosity and permeability data based on original logging data and classify and evaluate tight oil reservoirs through artificial rules and experience [17]. However, in the context of today's growing emphasis on clean and sustainable energy, machine learning stands out as a promising avenue for advancing tight oil reservoir evaluation studies [10,12,18]. Fan et al. developed a fracture prediction model based on quantitative multi-parameter regression, constrained by the average density of typical drilling-induced fractures, using geological (core and imaging logging) and geophysical methods [19]. Machine learning algorithms can overcome these limitations by identifying complex relationships and patterns that are difficult to discover through manual processes. By applying machine learning algorithms to well-logging data, researchers can explore new rules, identify previously unknown patterns, and improve the accuracy and efficiency of reservoir characterization and evaluation [20]. After large volumes of well-logging data are analyzed, machine learning algorithms can automatically learn patterns and make predictions, improving the accuracy and efficiency of reservoir characterization and evaluation [13].

The evaluation of reservoir quality is an important task in petroleum exploration and production. The Reservoir Quality Index (RQI), combining porosity with permeability, is the usual parameter to evaluate reservoir pore structure. The RQI can provide a macro view of the reservoir quality and it is widely used [21]. The pores and throats in the reservoir formation can be modeled as spherical pores and tubular pores, respectively, and their configuration relationships can significantly affect the physical parameters of the reservoir [22]. Therefore, it is necessary to analyze the size distribution of the pore and throat to perform the reservoir evaluations efficiently. In order to obtain the parameters of pore structure to characterize the pore size distribution, Liu et al. (2013) [22] developed an optimization inversion of the spherical–tubular model. Numerous parameters can be obtained, including the geometric mean value of the T2 spectrum (T2lm), the separation coefficient of spherical holes (SPS), the separation coefficient of tubular holes (SPC), the mean radius of spherical holes (dms), and the mean radius of tubular holes (dmc). And the parameters can be used as a foundation for reservoir evaluation. The optimization inversion of the spherical-tube model proposed by Liu et al. (2020) [23] shows an effective way to analyze the pore structure and provides parameters for reservoir evaluation. The RQI combining with pore structure analysis can provide a comprehensive understanding of the reservoir and optimize reservoir development strategies [22–24].

Amidst the global pursuit of sustainable and cleaner energy sources, our research pioneers a novel approach to reservoir evaluation. We not only leverage the traditional Reservoir Quality Index (RQI) but also integrate pore geometry parameters derived from

the optimization inversion of the spherical–tubular model. This integration of multiple parameters offers a more comprehensive understanding of reservoir characteristics, facilitating the optimization of reservoir development strategies.

Moreover, our study advances conventional methodologies by incorporating machine learning techniques into the reservoir evaluation process. The selection of machine learning methods, including the Random Forest algorithm (RF), is based on the necessity to enhance the accuracy and efficiency of reservoir evaluation. We acknowledge the importance of explaining why these methods are chosen. Therefore, we aim to provide a comprehensive rationale for the selection of each method, particularly highlighting the advantages they offer in addressing the challenges of reservoir evaluation. Our dataset comprises over 7000 entries from five wells, with four wells dedicated to training the models and the remaining one utilized for predictive analysis. This meticulous arrangement enables us to discern the varying performance of different machine learning methods in the realm of reservoir evaluation, providing a thorough understanding of their effectiveness and applicability.

Through this innovative approach, we not only contribute to the ongoing discourse on sustainable energy development but also provide practical insights for optimizing reservoir exploration and production strategies in the context of evolving energy demands.

## 2. Method

This chapter discusses the machine learning techniques and reservoir classification methods employed in the article. Three machine learning methods were utilized in the study: the Random Forest algorithm, Support Vector Machine algorithm, and Extreme Gradient Boosting algorithm. The article employed reservoir quality factors and optimized inversion methods by the spherical-tubular Model of NMR Logging for reservoir classification. The schematic of the workflow used in this work is shown in Figure 1.

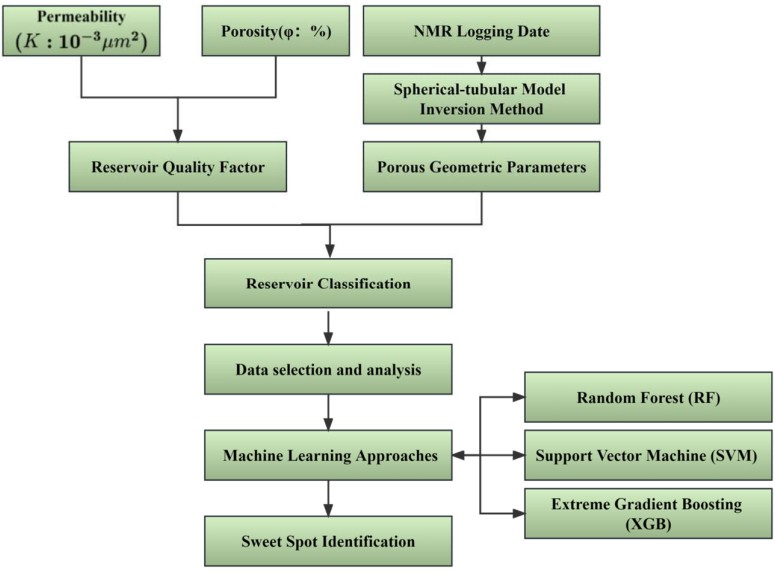

**Figure 1.** Schematic of the workflow presented in this work.

### 2.1. Machine Learning Methods

Machine learning methods encompass a set of algorithms designed to construct models that learn and improve autonomously based on data and statistical principles. These methods exploit patterns and regularities in the data, thereby enabling computers to perform tasks such as prediction, classification, clustering, and optimization. This paper uses Random Forest algorithm (RF), Support Vector Machine algorithm (SVM) and Extreme Gradient Boosting algorithm (XGB) for reservoir sweet spot prediction.

2.1.1. Random Forest Algorithm

The Random Forest algorithm, or RF for short, is a powerful and widely used supervised learning algorithm that can address both regression and classification problems. As an ensemble learning algorithm, RF integrates multiple decision trees to produce a final prediction result. In classification problems, RF determines the output class by taking the mode of the individual tree outputs. In regression problems, on the other hand, it uses the average output of each decision tree to obtain the final regression result. RF allocates samples to each tree by randomly drawing from the dataset and replacing the drawn samples. With each data extraction, a decision tree model is built, and ultimately, all the decision trees are integrated to form a "forest". The final prediction is determined through a voting decision process. Unlike the decision tree algorithm, RF introduces two distinct random conditions: (1) extract training datasets randomly from the entire dataset, and each extraction is attributed to be a decision tree; (2) select a subset of feature attributes randomly from the datasets which are attributed into the extracted training dataset. These two random conditions enable RF to achieve better performance compared with a single decision tree. To address limitations including overfitting and high variance, Random Forest (RF), as a variant of the decision tree algorithm, is firstly introduced by Breiman (Breiman, 2001) [25]. Liaw and Wiener (2002) confirmed the effectiveness of RF on a range of datasets, showing that it outperformed other popular classification algorithms, such as support vector machines and artificial neural networks [26]. Since then, RF has become a favorable algorithm in many applications, including image classification, gene expression analysis, and credit scoring.

The algorithm of RF is a powerful and versatile supervised learning algorithm that combines the strengths of decision trees with ensemble learning. The algorithm has excellent strength to handle various problems, such as regression and classification problems, and data missing problems, and maintains high performance on datasets with a large number of variables. As a consequence, it is a valuable tool in many research works.

Figure 2 depicts the regression process of the RF, which involves the generation of multiple decision trees produced by the bootstrap sampling method with replacement and random feature selection. Although each decision tree is independent, each of them contribute to the prediction process. The final prediction of the Random Forest regression model is obtained by calculating the arithmetic mean of the predictions from each individual decision tree. This ensemble approach enhances the prediction accuracy and generalization performance of the model [12,26]. In addition to the number of decision trees, the maximum number of features in a single tree is also an important parameter that should be adjusted during the modeling process of the RF. Other key parameters include the minimum number of samples required to split a node and the minimum number of samples required to be at a leaf node. These parameters can be tuned to optimize the performance of the model for different applications.

There are three steps included in the algorithm of RF, and they can be performed as the following operations. First of all, the bootstrap resampling method is used to extract k samples from the original training set, so that the sample size of each sample is consistent with the original training set. Then, use the obtained k samples to build a decision tree model, respectively, to obtain k different classification results; the final result is obtained by arithmetically averaging the results of each decision tree.

RF constructs different training sets by randomly extracting samples from the original training sets. Accordingly, there will be differences in the generation of classification models using training sets, which can improve the classification performance of combined classification models as a whole. The k samples obtained from sampling are used to construct classification models, respectively. Each classifier will correspond to an output result and vote on all the results obtained. The final result is obtained by arithmetic averaging the results of each decision tree.

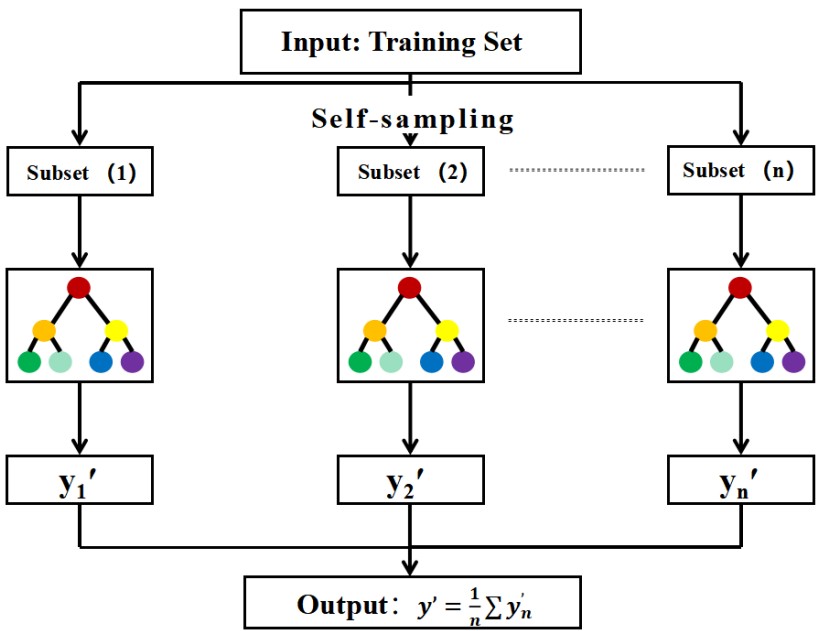

**Figure 2.** Schematic diagram of the RF.

The category with the most votes shall be regarded as the final classification result. The final classification decision is shown in Equation (1):

$$y' = \frac{1}{n}\sum y'_n \tag{1}$$

where, represents the output structure of the classification model and represents the result of a single decision tree.

Figure 3 depicts the usage of the Random Forest algorithm in the current study. RF is an integrated algorithm that combines multiple decision trees, and the final output of its regression is the average output of all tree numbers. Randomness mainly embodies two aspects: random selection of data and random selection of features. The random selection of data is to build a data subset from the returned sampling in the original data set and use the data subset to build a sub-decision tree. This data selection method is called Bootstrap sampling. Then, random feature selection is introduced in the training process, and the optimal feature is selected from these features. This operation generates a large number of decision trees, which are unrelated to each other and each of which participates in the judgment process. Different trees are good at choosing different features, that is, the input data can be judged from different angles. Finally, the results of each decision tree are summarized to jointly determine the final output to improve the diversity of the system, thus improving the accuracy of the prediction model. RF is often used in real analysis. Compared with a single decision tree, this method can easily reduce model errors and has better generalization performance. The random forest mainly adjusts two parameters: the number of decision trees and the maximum number of features in a single tree. Because it is not sensitive to outliers in the data set and does not require too much parameter tuning, the setting of hyperparameters will not fluctuate greatly for this method. Even if default parameters are used, better results can be achieved, and it is robust.

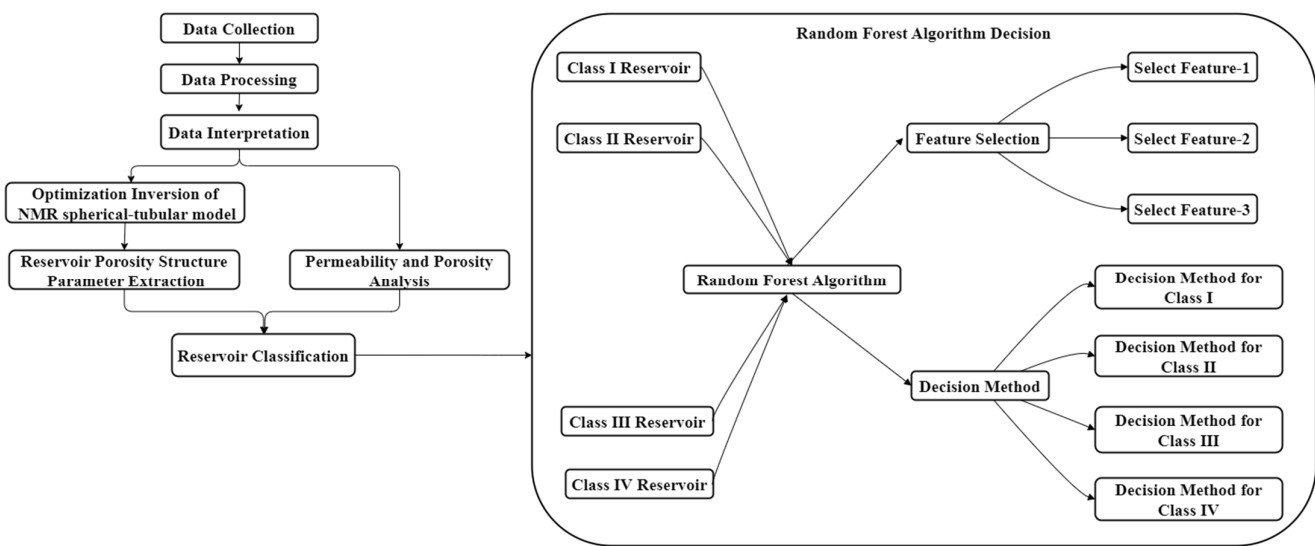

**Figure 3.** Application diagram of RF.

### 2.1.2. Support Vector Machine Algorithm

The algorithm of Support Vector Machine (SVM) is an important method in machine learning, specifically for tasks such as classification, regression, and anomaly detection [27]. The fundamental concept of SVM is to identify the optimal hyperplane that can effectively separate data points belonging to different classes while maximizing the margin between the closest points of the two classes. SVM is known for its high prediction accuracy, robustness, and generalization ability. Another advantage of the SVM algorithm is the ability to use kernel functions, which can transform non-linear problems into linear ones [28]. Linear, polynomial, and radial basis functions are the most commonly used kernel functions. The three-kernel functions are suitable for different types of data. The choice and parameters of the appropriate kernel function may have a significant influence on the performance of the SVM. Therefore, how to select an appropriate kernel function in the SVM modeling process is a key step.

SVM is a popular method in supervised learning, and it has been shown to be effective even with small sample sizes. SVM is initially proposed for classifications and its success is due to its ability to find the optimal hyperplane that maximizes the margin between the closest points in two different classes.

To expand on the concept of maximizing the margin, Figure 4 demonstrates that there are countless lines that can separate the data samples. However, only the one line with the maximum margin, which is represented by the distance between the two parallel dashed lines, will correctly divide the data. The points on this line are known as the support vectors and are the critical points used in determining the hyperplane. In practice, it is not always possible to find a hyperplane that perfectly separates the data, and the SVM may avoid some misclassifications to a certain degree through the use of a penalty parameter.

In the Figure 4, solid black circles and hollow black circles represent different types, while red circles represent points located on the boundary. The algorithm effectively distinguishes between different types of data.

The goal of SVM is to find a hyperplane of n-dimensional space (n is the number of features) that can classify data points. In the sample space, the partition hyperplane can be written in the form of a generalized vector, which is described by the following linear equation:

$$w^{\mathrm{T}}x + b = 0 \tag{2}$$

where x is the input vector, the vector in the sample set. W = (w1, w2, w3, ......, wd) is a normal vector, representing the direction of the hyperplane, and each vector is an adjustable weight vector. b is the intercept, also known as bias, and represents how

far the hyperplane is offset from the origin. Let us call this plane (w, b). According to the calculation formula from point to line, the distance between any point x in the sample space and the hyperplane (w, b) can be written as follows:

$$r = \frac{\left|w^T x + b\right|}{\|w\|} \tag{3}$$

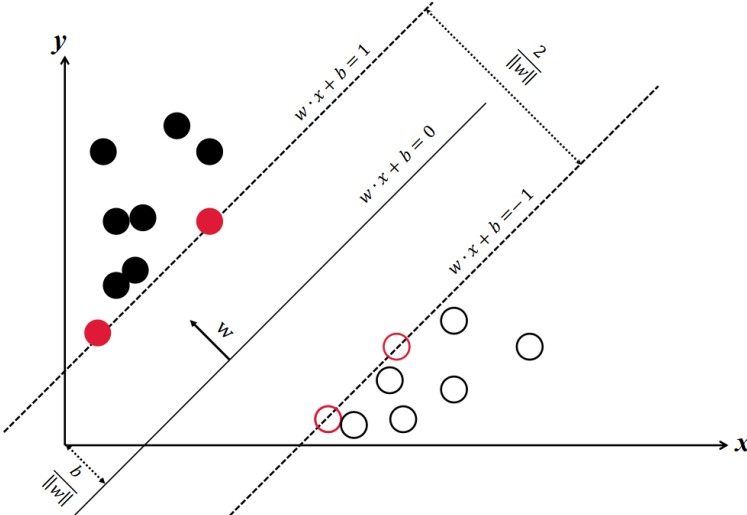

**Figure 4.** Schematic diagram of the SVM.

Suppose that the hyperplane (w, b) can correctly classify training samples, that is, for $(x_i, y_i) \in D$, if $y_i = 1$, $w^T x + b > 0$. If $y_i$ is equal to negative 1, $w^T x$ plus b is less than 0.

$$\begin{cases} w^T x + b \geq 1, \ y_i = 1 \\ w^T x + b \leq -1, \ y_i = -1 \end{cases} \tag{4}$$

As shown in the figure, the distance of a point from the hyperplane can be expressed as the degree of certainty or accuracy of classification prediction. The sample closest to the hyperplane makes the equal sign in the above equation true. These are the support vectors.

Figure 5 depicts the usage of the SVM algorithm in the current study. SVM is a powerful and widely used machine learning model. It can deal with linear classification problems, can deal with nonlinear classification problems and outlier detection. It is one of the most popular machine learning models, especially suited for complex classification problems with small to medium data sets.

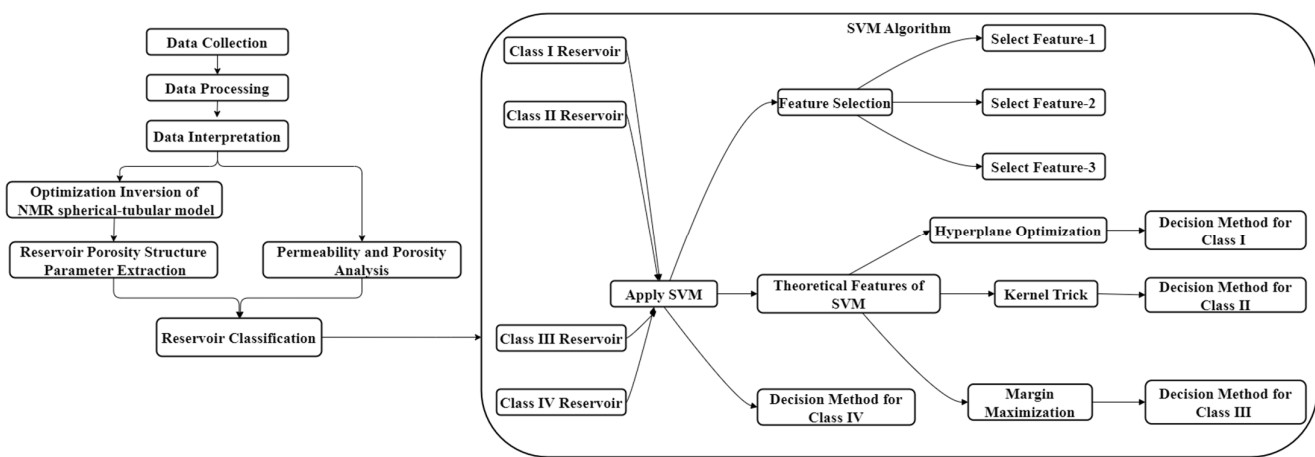

**Figure 5.** Application diagram of SVM.

### 2.1.3. Extreme Gradient Boosting Algorithm

Boosting is an ensemble learning algorithm that combines multiple weak learners to create a powerful model. It keeps iteratively training new models, and then focuses on the samples that were misclassified in the previous iterations. Boosting algorithms are divided into two main categories: gradient boosting and adaptive boosting. Extreme Gradient Boosting (XGB) is a type of gradient boosting algorithm that has been shown to outperform traditional gradient boosting techniques in many machine learning tasks. XGB sequentially combines base learners to improve the model's accuracy. The algorithm works by adding decision trees in each iteration to fit the residuals in the previous iteration's prediction (Figure 6). During the construction of new decision trees, XGB considers the importance of each feature to optimize the model effectively.

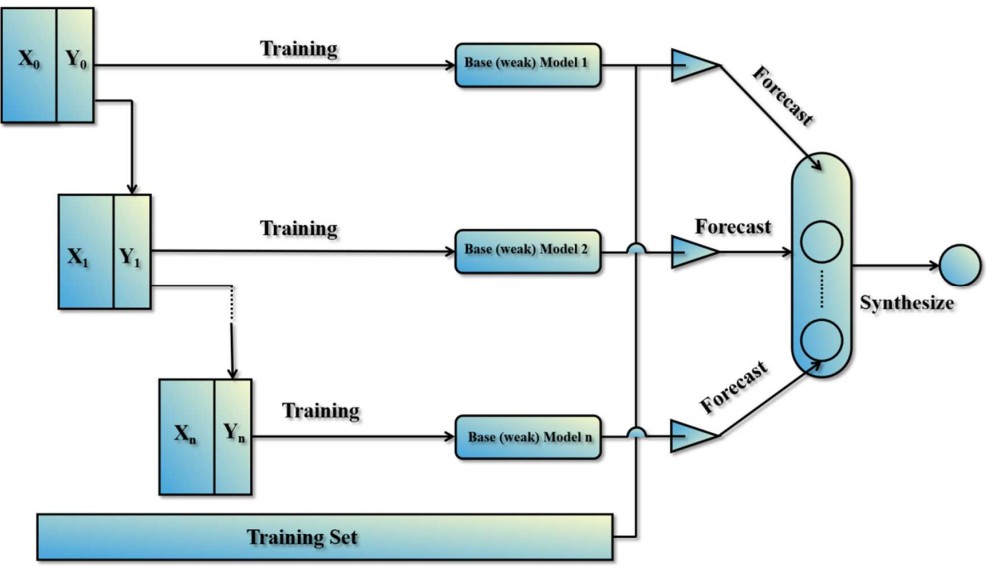

**Figure 6.** Schematic diagram of the XGB.

If a base learner makes imperfect predictions due to inherent algorithmic flaws, another base learner can be used to compensate for the "imperfect parts". The key principle of XGB is built up based on the rule above. By adding multiple base learners, the algorithm can continuously refine these "imperfect parts" and produce an ensemble model with excellent predictive accuracy and generalization performance.

Boosting is a popular ensemble learning technique, and the Gradient Boosting Decision Trees (GBDT) is a well-known example [29]. The GBDT algorithm trains a sequence of decision trees by fitting each tree to the residual errors left by the previous trees. In

this process, the overall model error is reduced, and the powerful model is then created. XGB, another boosting algorithm, was developed by Chen Tianqi and others as an open-source project to minimize model bias in supervised learning [30]. While the XGB is also a gradient boosting algorithm, it offers several improvements over the GBDT. For instance, the XGB employs a second-order Taylor expansion to calculate the objective error function (loss function), which enhances its ability to model complex relationships among variables. Additionally, the XGB introduces a regularization term in the loss function, which simplifies the model's computations and enhances its predictive accuracy and generalization performance [31].

Extreme gradient lifting algorithm XGB is a tree-boosting algorithm. Compared with the traditional gradient lifting decision tree algorithm, the XGB algorithm innovatively makes use of the second derivative information of loss function. This makes the XGB converge faster, ensures higher solving efficiency, and also increases expansibility. Because as long as a function meets the condition of the second derivative, this function can be used as a custom cost function under appropriate circumstances. Another advantage of the XGB is that it draws on the column sampling method of the RF, which further reduces the computation and overfitting. Currently, the widespread adoption of XGB stems not only from its model's impressive performance and rapid processing speed, enabling it to handle large-scale data computations, but also from its versatility in addressing both classification and regression problems effectively.

XGB algorithm can be expressed as follows:

$$\hat{y}_i = \sum_{k=1}^{K} f_k(x_i) \tag{5}$$

where K represents the number of trees and $f_k(x_i)$ represents the classification result of the i-th sample in the K-th tree.

As can be seen from the expression of the XGB, this model is a set of iterative residual trees, and one tree will be added in each iteration. Each tree will eventually form a model formed by the linear combination of K trees by learning the residual of the previous (K − 1) trees.

The XGB provides a number of metrics, including the total number of times each feature is used for splitting Fcount, the average gain of each feature, and the average coverage rate of samples after each feature splits nodes, ensuring the construction of a decision tree. The accuracy of node segmentation in the process makes the XGB have good performance.

For any tree whose structure is determined, there are the following:

$$Fcount = |C| \tag{6}$$

$$\overline{Gain} = \frac{\sum Gain_C}{Fcount} \tag{7}$$

$$\overline{Cover} = \frac{\sum Cover_C}{Fcount} \tag{8}$$

where C is the feature set used by all trees to generate nodes, $Gain_C$ is the gain value generated after each tree is divided by features in C, and $Cover_C$ is the number of samples falling on each node when the tree is divided by features in C.

XGB, which stands for Extreme Gradient Boosting, is a powerful machine learning algorithm widely used for both classification and regression tasks. This algorithm is particularly effective at handling structured data and is renowned for its exceptional predictive performance. The application diagram in Figure 7 likely outlines a specific use case or implementation of XGB in a particular context.

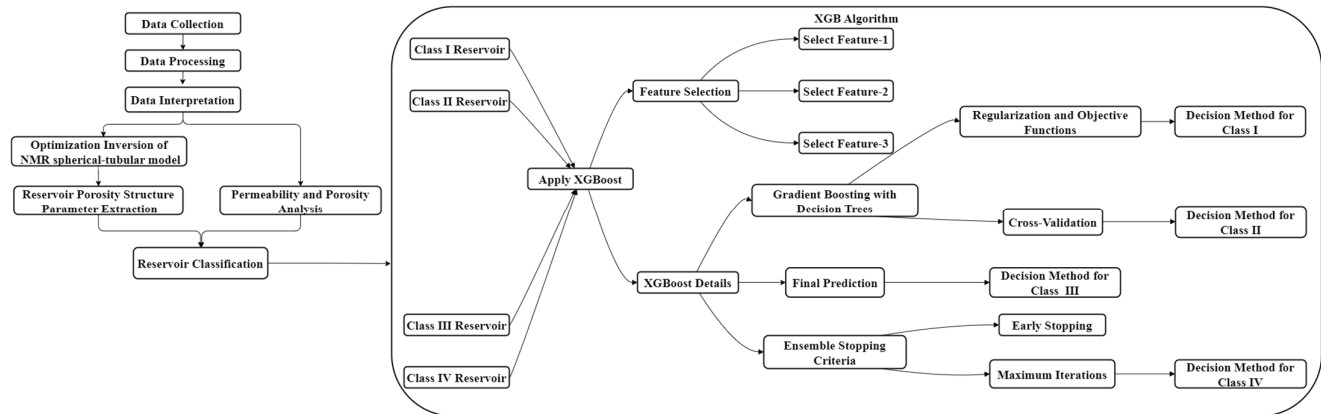

**Figure 7.** Application diagram of XGB.

Figure 7 depicts the usage of the XGB algorithm in the current study. In essence, XGB is an ensemble learning technique that combines the predictions of multiple weak models, typically decision trees, to create a strong and highly accurate model. It operates by iteratively building and optimizing these decision trees to minimize a specified objective function, such as mean squared error for regression or log-loss for classification.

### 2.2. Reservoir Classification Method

Reservoir classification is one of the important tasks in subsurface energy exploration and development. It aims to classify the underground reservoirs to provide valuable information about reservoir properties and hydrodynamics characteristics. In order to achieve accurate reservoir classification, researchers have proposed various reservoir classification methods. In this paper, the classification method of reservoir quality index and the optimization inversion method of the spherical-tube model are used to classify and evaluate the reservoir in the study area.

Classification by Reservoir Quality Index

The complex pore structure and strong reservoir heterogeneity make it challenging to accurately evaluate reservoirs based on simple parameters, such as porosity and permeability alone. The reservoir quality index (RQI) has been introduced to provide a more comprehensive assessment of reservoirs. Both porosity and permeability are the essential macro parameters for evaluating reservoirs, but they do not always provide a complete picture of the pore structure. The RQI is a macro parameter that combines porosity with permeability to provide a more accurate representation of the pore structure and reservoir quality. The RQI approach has been widely adopted in petrophysical classification and reservoir characterization [21,32,33]. The micro pore-throat structure can be also characterized in the RQI, which has been used to identify the complex pore structure and pore heterogeneity in reservoir evaluations.

Define the reservoir quality factor RQI as the following:

$$\text{RQI} = \sqrt{\frac{K}{\varnothing}} \tag{9}$$

where $\varnothing$ is the effective porosity, %; K is permeability, $10^{-3}$ um$^2$.

When combined with reservoir micro pore-throat structure parameters, the RQI facilitates smooth evaluations of the complete pore structure in reservoir classification through presenting the pore-throat structures and petrophysical properties within the reservoir [34,35]. The RQI serves as an effective method in petrophysical classification and a characteristic parameter reflecting the micro pore-throat structure. The higher the RQI, the better the micro pore-throat structure in the reservoir. Therefore, the RQI is a crucial tool in

reservoir evaluation and management, allowing for the identification of potential reservoirs for development and the optimization of production strategies.

The study area belongs to the clastic rock reservoir, which has developed a complex porous reservoir and percolation system. The reservoir has deep burial, a strong epigenetic process, a complex pore structure, and high immobilized water saturation, resulting in little difference in the logging response between the oil and water layers. According to the method of the ROI reservoir quality factor, the reservoirs in the study area are classified as shown in Table 1 below.

**Table 1.** Classification basis of reservoir.

| Reservoir Classification | Reservoir Quality Index (RQI) | Permeability K ($10^{-3}$ $um^2$) | Porosity $\varphi$ (%) |
|---|---|---|---|
| I | >0.62 | >5 | 9–21 |
| II | 0.23–0.62 | 0.5–5 | 8–17 |
| III | 0.1–0.23 | 0.07–0.5 | 3–14 |
| IV | <0.1 | <0.07 | 2–13 |

*2.3. Optimization Inversion of NMR Spherical–Tubular Model*

The pore structures in real reservoir rocks are too complex to characterize them using any analytical methods. However, if reasonable approximation conditions are set up, specific models can be employed to approximate the pore structure of rocks. In this study, the sphere-tubular model is utilized to classify and categorize reservoir rocks. Provided that the rock pores can be approximated by the combination of tubular pore and spherical pore, the spherical–tubular model is used to perform our research in this paper. By analyzing the parameters of the sphere-tubular model, the characteristics of the rock pore structure can be estimated to provide valuable references for the evaluation and development of subsurface reservoirs.

The spherical–tubular model is a useful tool to approximate the complex pore structures embedded in real reservoir rocks. This model is based on the idea that rock pores can be approximated as a combination of spherical pores (the pore part) and tubular pores (the throat part) [36]. The different matching ways between spherical pores and tubular pores represent the different types of pore structures. The model assumes that pores are sorted by their volume size and that each group contains a set of spherical–tubular models with identical shapes. The spherical–tubular models in different groups have similar shapes, but their different radii of the tubular pore and spherical pore exhibit different configurations, which represent the different numerical relationships between the radii of the tubular pore and spherical pore (Figure 8). Not only is the spherical–tubular model a perfect representation of the actual pore structures embedded in rocks, but also it provides a useful approximation to understand and better classify the different types of reservoir rock.

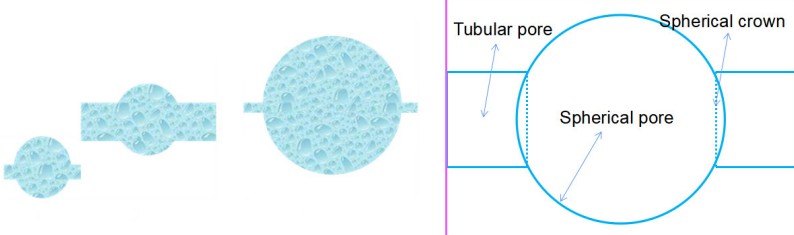

**Figure 8.** Simulation and parameter description of the spherical–tubular model.

After performing the spherical–tubular model optimization inversion, a mapping relationship between the parameters of the spherical–tubular model and core nuclear magnetic resonance (NMR) data or NMR logging data is established, as suggested by [22,36].

The pore shapes within the rock are assumed to be sorted in terms of their volume size, and each pore component contains different spherical–tubular models. The data inversion of NMR echo is performed in the transverse time distribution of NMR; the parameters of the spherical–tubular model can be determined using Equations (10) and (11).

$$T_{2i} = f(R_s, C_d, R_e) \tag{10}$$

$$C_d = \frac{R_c}{R_s} \tag{11}$$

where, $T_{2i}$ is the i-th distribution point value of echo signal inversion, ms;

$R_s$, $R_c$, $R_e$, are the spherical pore radius, tubular pore radius and equivalent spherical pore radius in um, respectively.

$C_d$ is the radius ratio of the tubular pore to the spherical pore, and it is dimensionless.

The optimization inversion based on the sphere–tube model is a powerful algorithm to understand the pore-throat structure in reservoirs. By conducting the optimization inversion, a great number of parameters can be obtained, which includes the optimized inversion T2 spectrum (ms), T2 spectrum of spherical pore (T2S, ms), T2 spectrum of tubular pore (T2C, ms), geometric mean of T2 spectrum (T2lm, ms), sorting coefficient of spherical pores (SPS, dimensionless), sorting coefficient of tubular pores (SPC, dimensionless), mean radius of spherical pore (dms, um), and mean radius of tubular pore (dmc, um), etc. The distributions and combinations of spherical and tubular pores are described by the parameters in reservoirs from different perspectives, allowing for a more comprehensive understanding of the pore-throat structure.

In order to evaluate the reservoir, we selected the parameters derived from the optimized inversion to conduct our research. The parameters include the geometric mean of the T2 spectrum (T2lm, ms), sorting coefficient of spherical pores (SPS, dimensionless), sorting coefficient of tubular pores (SPC, dimensionless), mean radius of spherical pores (dms, um), and mean radius of tubular pores (dmc, um), and they provide a comprehensive evaluation of the reservoir pore-throat structure from different perspectives.

## 3. Data Selection and Analysis

### 3.1. Data Preprocessing

Effective data preprocessing is essential to build an accurate and reliable pore-permeability model using machine learning techniques. This job involves several critical steps, including data feature analysis, data clustering analysis, data standardization, and data set partitioning. By performing these steps, we can ensure that the data is appropriately prepared and organized for the subsequent modeling process, which can lead to more accurate and effective predictions [37].

The data used in this study are sourced from the Bohai Basin of China, which is known for its complex pore reservoir, deep burial of flow systems, and strong diagenesis, resulting in a complex pore structure. This area is characterized by tight sandstone lithology. The selected date set comprises post-logging data, including the geometric mean of T2 spectrum (T2lm, ms), the sorting coefficient of spherical pore (SPS, dimensionless), the sorting coefficient of tubular pore (SPC, dimensionless), mean radius of spherical pore (dms, um), mean radius of tubular pore (dmc, um), permeability (K, $10^{-3}$ um$^2$), porosity (POR, %), and reservoir quality factor (RQI, dimensionless).

In this study, the reservoirs were divided into four categories based on the reservoir classification presented in Table 1. Both the distributions of porosity and permeability of the four categories in the reservoirs are shown in Figure 9, and the four categories are clearly distinguished. The studies in the paper suggest that the proposed classification is effective in differentiating the storage and flow characteristics of the reservoirs.

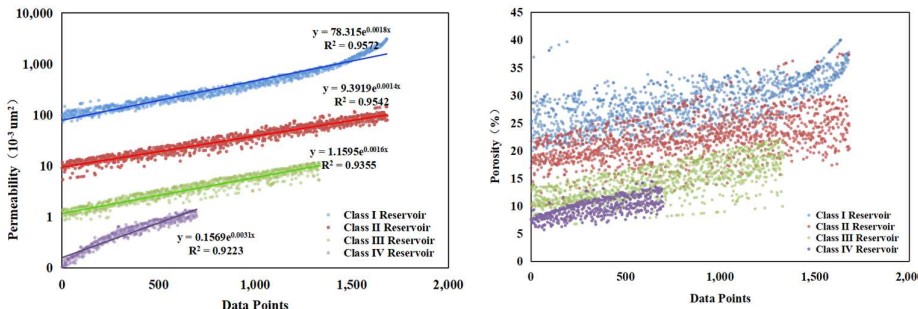

**Figure 9.** Analyses of reservoir porosity and permeability characteristics.

Furthermore, the findings of this study demonstrate the potential of reservoir classification to enhance reservoir characterization and management. By accurately categorizing reservoirs based on their storage and flow characteristics, petroleum engineers can develop more targeted zones and effective management strategies to optimize production and maximize recovery.

Generally, the applicability of a model to new samples is judged by its generalization performance. It is necessary to establish a "test data set" to test the generalization performance for the model and convert the "test error" of the test data into a generalization error. The test samples are usually extracted from the sample data and follow the principle of avoiding test samples as much as possible from appearing in the training set. Currently, there are three main ways to perform the model evaluation, namely cross-validation, holdout and bootstrapping.

### 3.1.1. Setting Aside Method

The "set-aside method" is a technique to partition a data set D into two subsets, a training set (S) and a test set (T), to assess the effectiveness of a model. During the partitioning process, D is divided into S∪T, where S and T are disjoint sets. In the model-building process, the training set (S) is used for training, and once completed, the estimation accuracy of the model is verified by the test set, which is used to estimate the model's generalization error.

### 3.1.2. Cross-Validation Method

In the "cross-validation method", all the data in dataset (D) is first divided into k mutually exclusive subsets in a similar size, namely D = D1∪D2∪D3∪...∪Dk, where Di∩Dj = ∅. Each subset Di tries to maintain the consistency of the data distribution that is obtained from D through stratified sampling. Then, the union of K − 1 subsets is taken as the training set, and the remaining subset is taken as the test set. Both of the subsets are designed to train and test the model, respectively. The stability and truthfulness of the cross-validation method in evaluating the model largely depends on the value of k [38,39].

Similar to the setting aside method, different methods can be used to partition the training set and test set based on the characteristics of the samples in the dataset. In order to optimize the model dataset, multiple modeling and averaging are often used to verify the effectiveness of the model according to the subset number k of the dataset partition. Typically, k is set to 10.

### 3.1.3. Bootstrap Method

In order to mitigate the impact of varying training sample sizes and improve experimental estimation efficiency, researchers have proposed the "bootstrap sampling method", commonly known as the "bootstrap method" [40]. Given a dataset A containing *n* samples, a dataset A′ is generated by *n* iterations of sampling with replacement. This method may result in the occurrence of repeated under-sampling during the data set extraction process.

Using simple estimation, the probability of a single sample not being selected during $n$ iterations of random sample data is approximately 0.368 (as shown in Equation (12)).

$$\lim_{n \to \infty} \left(1 - \frac{1}{n}\right)^n = \frac{1}{e} \approx 0.368 \tag{12}$$

Therefore, it can be observed that 36.8% of the samples are left out during the process of data sample extraction with this method. However, when dealing with small sample data, it becomes challenging to establish an effective training/test dataset using the bootstrap method. On the other hand, this method can generate multiple independent training datasets during the data partitioning process, which is beneficial for ensemble learning methods. Nevertheless, the drawback of the bootstrap method is that the generated datasets change the distribution of the initial dataset, leading to estimation bias. Hence, when dealing with sufficiently large sample data, the setting aside method and cross-validation method are typically preferred.

In this study, in order to verify the accuracy of the machine learning method for reservoir prediction, we classified the required data by ourselves. In total, we selected five wells with over 7000 sets of data. We chose four wells as the training data and the remaining one as the prediction data. Then, we used three machine learning algorithms, namely SVM, RF, and XGB, to perform the model training and prediction.

*3.2. Data Correlation Analysis*

Correlation analysis is a statistical technique used to analyze the degree of association between one variable and another variable. In order to visually represent the correlation results, in this study, the heatmap function in the Seaborn package of the Python programming language is utilized.

The Pearson correlation coefficient represents the agreed meaning of the covariance between two variables and the standard deviation difference. It ranges from $-1$ to $1$. The coefficient of $1$ indicates a perfect positive correlation, while the coefficient of $-1$ indicates a perfect negative correlation, and the coefficient of $0$ indicates no correlation. The specific formula to calculate the Pearson correlation coefficient is presented as follows (Equation (13)):

$$\rho_{x,y} = \frac{\text{Cov}(x, y)}{\sqrt{\text{Var}(x)\text{Var}(y)}} \tag{13}$$

where the $\text{Cov}(x, y)$ is the covariance of x and y; the $\text{Var}(x)$ and $\text{Var}(y)$ are the variances of x and y, respectively.

Because the correlation analysis enables us to understand the relationship between various rock physical parameters and provides the foundation for subsequent modeling and prediction, it is a critical step in reservoir intelligent evaluation and prediction. In this study, the Pearson correlation coefficient was used to measure the correlation between different parameters. The range of the coefficient is from $-1$ to $1$, which indicates the strength and direction of the linear relationship between the two parameters. The coefficient close to $1$ suggests a strong positive correlation between the two feature vectors, while the coefficient close to $-1$ indicates a strong negative correlation. A coefficient close to $0$ indicates a low correlation between two features.

For instance, the high correlation coefficient between porosity and permeability indicates a close linear relationship between the two parameters, while the low correlation coefficient between porosity and depth suggests that the linear relationship between the two features is relatively weak. By using the heat-map visualization tool, we can easily identify groups of features with high correlation and understand the relationship between different rock physical parameters. Figure 10 shows the results obtained from data visualization combined with the heatmap function. The heat-map provides an intuitive way to visualize the correlation between different parameters, which can aid the group identification of highly

correlated features. By examining the heatmap, we can gain insights into the underlying relationships among features and improve the accuracy of predictive models.

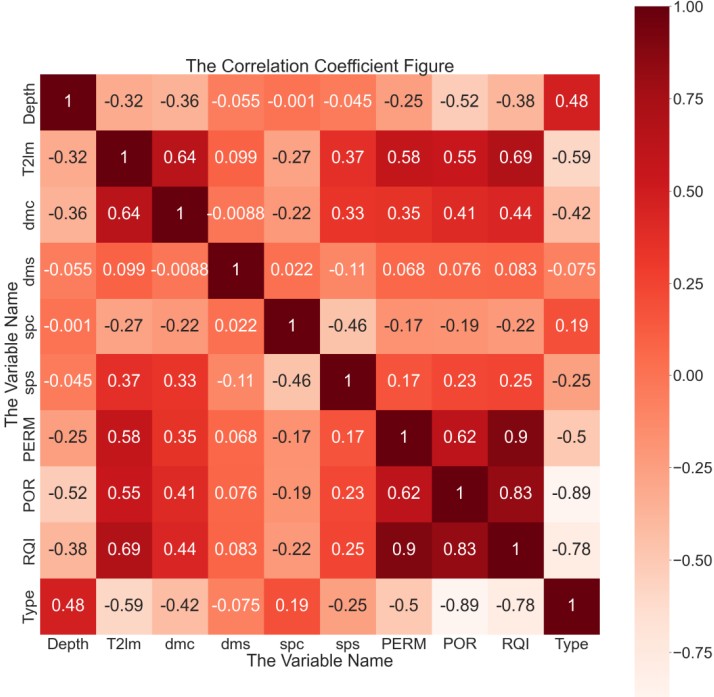

**Figure 10.** Correlation analysis diagram.

Figure 10 reveals several key findings related to the interrelationship among the different rock physical parameters. First, the geometric mean of T2 spectrum (T2lm, ms) is strongly correlated with permeability (K, $10^{-3}$ um$^2$), porosity (POR, %), reservoir quality factor (RQI, dimensionless), and the mean radius of tubular pores (dmc, um). Second, permeability (K, $10^{-3}$ um$^2$) has a significant correlation with the geometric mean of T2 spectrum, porosity, and reservoir quality factors. Third, the reservoir type is well-correlated with most of the parameters, except the mean radius of the spherical pore (dms, um).

By identifying the relationships among different rock physical parameters, we can gain insights into the reservoir properties and optimize production strategies. The results presented in Figure 10 suggest that some parameters are more strongly correlated than others, highlighting the potential of predictive modeling based on these parameters.

## 4. Applications

Reservoir prediction analysis is a critical step in reservoir intelligent evaluation and prediction. In this study, machine learning algorithms were utilized for intelligent reservoir evaluation and prediction. In order to verify the effectiveness of this technology, we selected the xx oilfield reservoirs as the research area and carried out intelligent evaluation and prediction. By leveraging machine learning algorithms, we can gain deeper insights into the reservoir properties and optimize production strategies. These algorithms are designed to identify patterns and relationships within complex datasets, enabling us to make more accurate predictions about reservoir behavior. In this study, in order to verify the accuracy of the machine learning method for reservoir prediction, we classified the required data ourselves. In this paper, a total of five wells were selected, with a total of more than 7000 sets of data. We selected four of them as training data, and the remaining one for prediction.

Figure 11 shows the confusion matrix and accuracy graph of the prediction. It is evident from Figure 11 that all the three prediction methods performed well, with an accuracy greater than 90% in predicting the behavior of the reservoir. Notably, the Extreme Gradient Boosting algorithm achieved an accuracy of up to 97%, demonstrating its effectiveness

in evaluating and predicting reservoirs. Through the application of prediction analysis to the reservoir, we can understand the nature and distribution of the reservoir entirely, which provides valuable support for subsurface energy exploration and development. Furthermore, this technology can be utilized in petroleum prospects to offer reference and inspiration for other fields if the prediction analysis is required.

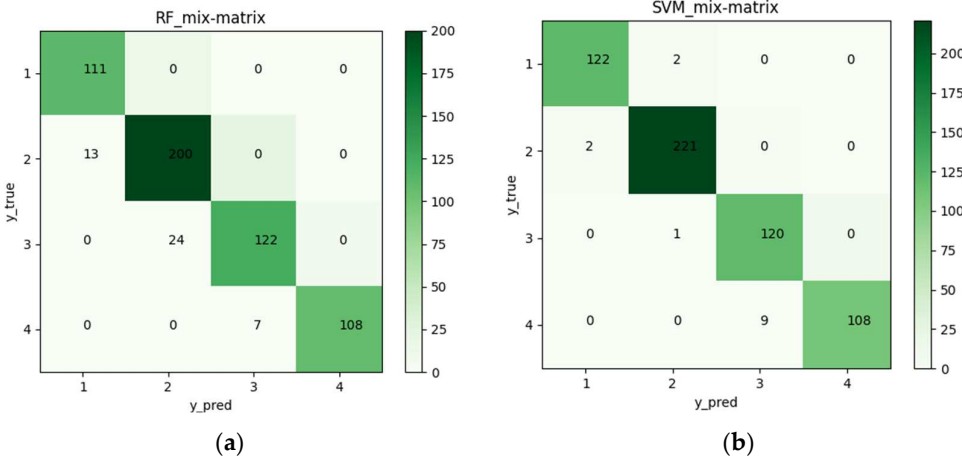

(**a**)　　　　　　　　　　　　　(**b**)

**Figure 11.** *Cont.*

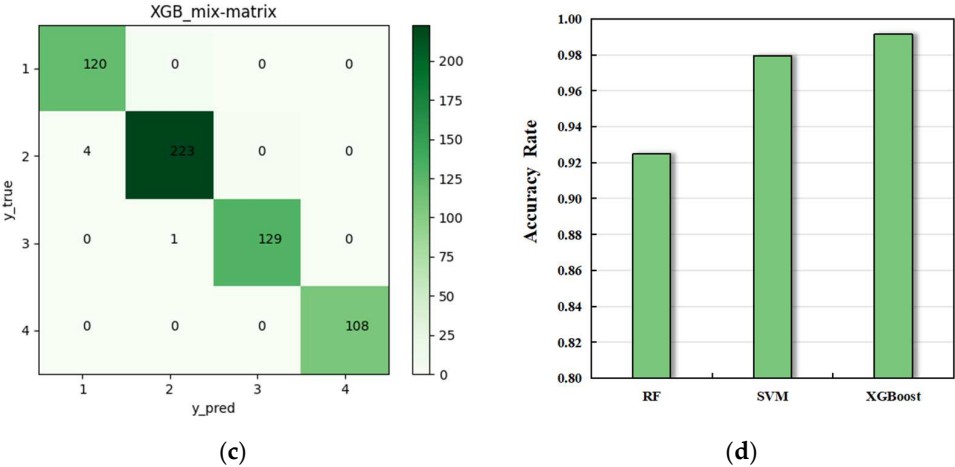

(**c**)　　　　　　　　　　　　　(**d**)

**Figure 11.** Confusion matrix and accuracy graph. (**a**) Prediction of confusion matrix by RF; (**b**) Prediction of confusion matrix by SVM; (**c**) Prediction of confusion matrix by XGB; (**d**) Different algorithm accuracy graph.

In order to understand the predictions of the three models better for the different reservoir types, the predictions were presented in the form of histograms for each model, as shown in Figure 12. Histograms provide a visual representation of the distribution of the predicted values for each reservoir type, enabling us to compare the performances of the three models across different types of reservoirs. By examining the histograms, we can gain insight into the effectiveness of each model in predicting the behavior of different reservoir types, which can inform the subsequent modeling and prediction efforts.

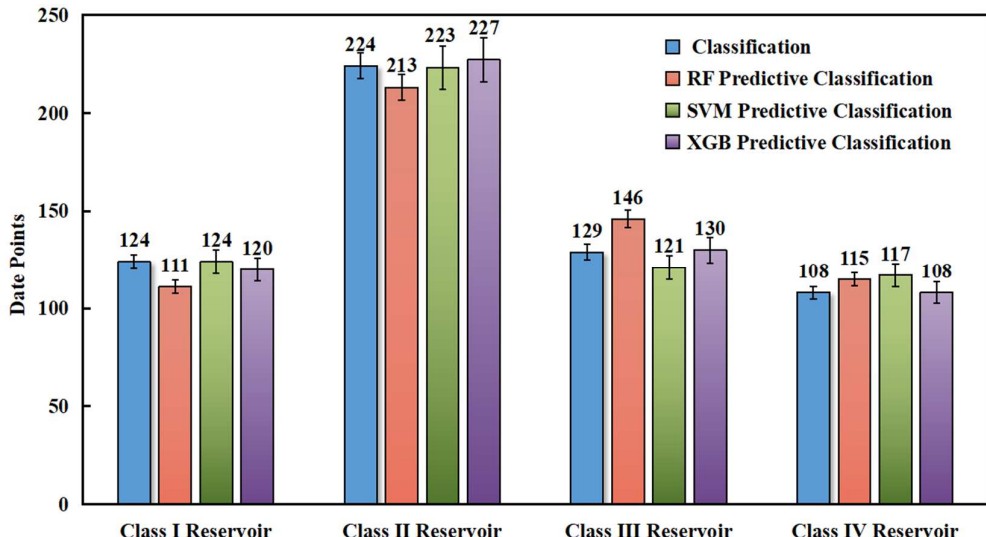

**Figure 12.** Prediction results of different prediction models for different reservoirs.

## 5. Discussion

In this study, the selected prediction data and well log data are plotted and analyzed. Among the three algorithms, the extreme learning algorithm demonstrates the highest accuracy. This is evident from the reservoir prediction classification in the figure, where the predictions derived from the XGB are chosen, as illustrated in Figure 13.

Figure 12 demonstrates that the SVM model achieved a prediction accuracy of 100% for the Class I reservoir, while the Extreme Gradient Boosting model achieved a prediction accuracy of 100% for the Class IV reservoir. When combined with the correlation analysis of previous reservoir classification and prediction parameters, it is clear that high porosity and high permeability exist in the Class I reservoir, and the porosity and permeability show a pretty strong relationship resulting in clear reservoir characteristics. In contrast, the Class IV reservoir is divided into the class of low porosity and low permeability, and the relationship between porosity and permeability seems to be weak, making comprehensive essential analysis if necessary. The Extreme Gradient Boosting algorithm model is better suited for comprehensive reservoir analysis, and it demonstrates lower prediction errors when compared to other models. By leveraging the strengths of this algorithm and examining all given training parameters, we can gain deeper insights into the reservoir behavior and optimize production strategies to maximize yields.

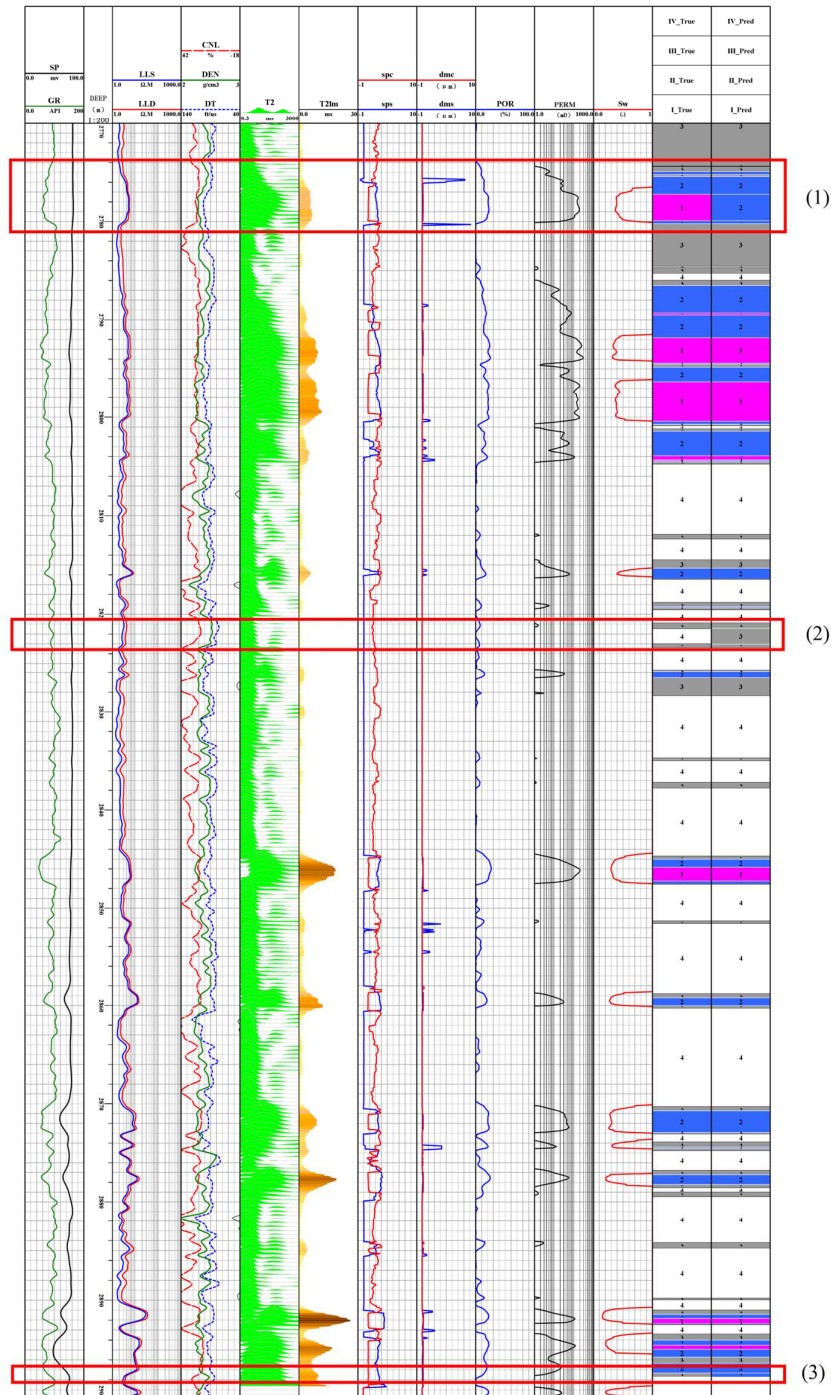

**Figure 13.** Predicted results of the well-log plot.

In Figure 13, the three different reservoirs are depicted in both the predicted and actual classifications:

(1)  The actual classification of the first member is Class I, while the predicted classification is Class II. The analysis of reservoir parameters reveals that high porosity, high permeability and high saturation are present in the reservoir. According to the optimization inversions of the spherical–tubular model, the geometric mean T2lm (T2) falls within the mid-range across the entire interval. Based on the four parameters, the sorting coefficient of the spherical pore (SPS, dimensionless), the sorting coefficient of the tubular pore (SPC, dimensionless), the mean radius of spherical pore (dms, μm) and the mean radius of tubular pore (dmc, μm), the development of spherical pore is superior to that

of tubular pore, indicating that the reservoir has favorable properties. However, due to the weak development of tubular pores, the permeability of the reservoir segment is limited. The prediction model of the extreme learning algorithm integrates seven parameter factors, leading to the prediction that the reservoir segment belongs to the Class II reservoir.

(2) The actual class of the second member reservoir is class IV, but the prediction class is class III. By analyzing the reservoir parameters, it can be seen that the porosity, permeability and saturation in the reservoir in this section shows poor relationship. According to the optimization inversions of the spherical–tubular model, the low geometric mean of T2lm exists in the section. After checking the sorting coefficient of the spherical pore (SPS, dimensionless), the sorting coefficient of the tubular pore (SPC, dimensionless), the mean radius of the spherical pore (dms, um) and the mean radius of the tubular pore (dmc, um), it shows that the development of both the spherical pore and tubular pore in this section is pretty poor. The prediction model of the ultimate lifting algorithm is based on the seven parameters in comprehensive consideration, so the prediction class of the second reservoir member is divided into class III with a certain deviation.

(3) The third member of the reservoir is partitioned into class III in the actual class, but it is divided into class II by the prediction model. After checking the parameters in the third member of the reservoir, all the parameters, such as the porosity, permeability, and saturation, seem to be low and poor. The greater geometric mean of T2lm (T2) is derived from the optimization inversion of the spherical–tubular model, and it is pretty great. The sorting coefficient of the spherical pore (SPS, dimensionless), the sorting coefficient of the tubular pore (SPC, dimensionless), the mean radius of the spherical pore (dms, um) and the mean radius of the tubular pore (dmc, um) are analyzed in these studies. All the parameters show that the development of spherical pores and tubular pores in this section is quiet weak. The prediction model of the ultimate lifting algorithm is based on the seven parameters in comprehensive consideration. Therefore, this reservoir segment is divided into class II by the prediction model with a certain deviation.

In summary, the prediction model of the XGB algorithm integrates seven factors to predict reservoirs. Although predicted deviations may occur in individual reservoirs, the overall accuracy is relatively high. The main reason for this incorrect partition is the lack of distinct characteristic parameters in the reservoir.

Accurate reservoir prediction holds significant importance for exploitation and utilization of underground resources. Precise forecasting of reservoir properties and pore structure can yield accurate reserve estimations, which in turn enable more efficient management and utilization of geological resources. In addition, reasonable reservoir classification can determine the optimum construction locations and determine stimulation methods to maximize resource recovery. The reservoir assessment can also assist us in identifying possible environmental risks, such as gas leakage and formation subsidence that may be caused by injection and production, thus ensuring risk prevention and sustainable development.

## 6. Conclusions

In this study, we have developed a reservoir classification approach by combining the NMR-based pore structure, reservoir quality factor, and machine learning algorithm. Thereinto, pore structure can be characterized by the optimized inversion of the spherical–tubular model based on NMR logging. We make the following conclusions:

(1) All the machine learning algorithms including Random Forest (RF), Support Vector Machine (SVM), and Extreme Gradient Boosting (XGB), can achieve predictive accuracies of more than 90%, and XGB has the highest accuracy of up to 97%.

(2) The results demonstrate that the pore structure information can be fully extracted by combining the reservoir quality factor with the porous geometric parameters obtained

from the optimized inversion of the spherical–tubular model, which highly correlates with the porosity and saturation of the target reservoir.

(3) In practical log interpretation applications, the XGB model is employed to predict reservoir properties. This model amalgamates reservoir quality factors with pore geometry parameters, which are derived from the optimized inversion of the NMR spherical-tube model. Although the overall predictive accuracy of the model is impressive, certain discrepancies are noted in the prediction of individual reservoirs. These deviations primarily stem from the indistinct characteristics of geometric structure parameters within the reservoir.

In conclusion, the approach of reservoir classification in this study can significantly improve the accuracy and efficiency of reservoir evaluation, providing valuable support for exploration and utilization of geological resources. As the world increasingly shifts its focus towards clean and sustainable energy sources, the optimization of conventional energy extraction, geological storage of sustainable energy and greenhouse gas becomes even more critical. The reservoir assessment with combining intelligent technology and low-cost logging data can guide the rational utilization and exploitation of underground resources, ensure the risk-free engineering activities and environment, thus giving insights for sustainable development.

**Author Contributions:** Conceptualization, W.Z. and T.L.; methodology, W.Z. and T.L.; software, W.Z.; validation, J.Y. and Z.Z.; formal analysis, W.Z.; investigation, W.Z.; resources, T.L.; data curation, W.Z.; writing—original draft preparation, W.Z.; writing—review and editing, J.T.; visualization, W.Z.; supervision, C.F. and J.T.; project administration, T.L.; funding acquisition, T.L. All authors have read and agreed to the published version of the manuscript.

**Funding:** This work is supported by the National 13th 5-Year Plan of Oil and Gas Program of China (2017ZX05019-001), CNPC Research Project (No. 2017 F-16), National Natural Science Foundation of China (No. 41476027), PetroChina Key Technological Program (No. 2016E-0503), the PetroChina Science and Technology Innovation Foundation (2021 DQ02-0501) and the Natural Science Foundation of Xinjiang Uygur Autonomous Region (No. 2021D01E22). All the research support funds are greatly appreciated.

**Data Availability Statement:** Dataset available on request from the authors: The raw data supporting the conclusions of this article will be made available by the authors on request.

**Conflicts of Interest:** Author Jian Yang was employed by Engineering Technology Research Institute of Southwest Oil & Gas Field Company, PetroChina. The remaining authors declare that the research was conducted in the absence of any commercial or financial relationships that could be construed as a potential conflict of interest.

## Nomenclature

| | |
|---|---|
| $y\prime$ | the output structure of the classification model. |
| $y\prime_n$ | the result of a single decision tree. |
| x | the input vector, the vector in the sample set. |
| W = (w1, w2, w3, ......, wd) | the normal vector. |
| b | the intercept. |
| K | the number of trees. |
| $f_k(x_i)$ | the classification result of the i-th sample in the K-th tree. |
| Fcount | the total number of times each feature. |
| $\overline{\text{Cover}}$ | the average coverage rate. |
| $\overline{\text{Gain}}$ | the average gain of each feature. |
| C | the feature set used by all trees to generate nodes. |
| $\text{Gain}_C$ | the gain value generated after each tree isdivided by features in C. |
| $\text{Cover}_C$ | the number of samples falling on each node when the tree is divided by features in C. |
| $\varnothing$ | the effective porosity, %. |
| K | the permeability, $10^{-3}$ um$^2$. |

| | |
|---|---|
| $T_{2i}$ | the i-th distribution point value of echo signal inversion, ms. |
| $R_s$ | the spherical pore radius, um. |
| $R_c$ | the tubular pore radius, um. |
| $R_e$ | the equivalent spherical pore radius in um. |
| $C_d$ | the radius ratio of the tubular pore to spherical pore, dimensionless. |
| $Cov(x, y)$ | the covariance of x and y. |
| $Var(x)$ | the variances of x. |
| $Var(y)$ | the variances of y. |

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
