# Peer review of "Approaches of Combining Machine Learning with NMR-Based Pore Structure Characterization for Reservoir Evaluation"

_sustainability, doi:10.3390/su16072774_

Round 1
Reviewer 1 Report
Comments and Suggestions for Authors
Tight sandstone gas is an important oil and gas resource, and the proportion of natural gas in our country is very large. However, the complexity and heterogeneity of tight reservoirs pose challenges in accurately identifying target layers by using traditional well-logging techniques. Machine learning may hold the key to solving this problem, as it enables computers to learn without being explicitly programmed and manually added rules. The machine learning method established in this paper has guiding significance for tight reservoir logging evaluation! However, there are some major problems with this article!
1.The application of machine learning to reservoir evaluation is a hot research issue in recent years. In the introduction, the author should strengthen the progress of machine learning in logging evaluation and put forward the innovation of this paper!
2.The author spends a lot of time studying the machine learning method in this paper, but the author neglects the basic geological understanding. Therefore, I suggest adding a section on geological background to analyze the object of this article in detail!
3.The reservoir classification is so simple that I find it hard to believe the credibility of this study!
4.Necessary experimental samples and instruments need to be added, such as nuclear magnetic resonance, rock characteristics, etc.
5.Which region does Figure 13 come from, and does this research method apply to all regions?
again, The quality of Figure 13 is very poor.
6.In general, I only saw the introduction of the method in this paper, and did not get the credibility of the method in geological research!
Comments on the Quality of English LanguageModerate editing of English language required
Author Response
Comments and Suggestions for Authors
Review1:
Tight sandstone gas is an important oil and gas resource, and the proportion of natural gas in our country is very large. However, the complexity and heterogeneity of tight reservoirs pose challenges in accurately identifying target layers by using traditional well-logging techniques. Machine learning may hold the key to solving this problem, as it enables computers to learn without being explicitly programmed and manually added rules. The machine learning method established in this paper has guiding significance for tight reservoir logging evaluation! However, there are some major problems with this article!
Q1.The application of machine learning to reservoir evaluation is a hot research issue in recent years. In the introduction, the author should strengthen the progress of machine learning in logging evaluation and put forward the innovation of this paper!
A:Thank you for your comments. We have incorporated the suggested modifications, including an emphasis on the innovative integration of traditional reservoir evaluation parameters with machine learning techniques. These additions aim to provide a clearer understanding of the novel contributions of our research.
2.The author spends a lot of time studying the machine learning method in this paper, but the author neglects the basic geological understanding. Therefore, I suggest adding a section on geological background to analyze the object of this article in detail!
A:Thank you for your comments. We have added a section providing detailed geological background information in the manuscript to address this concern. This new section aims to enhance the readers' understanding of the study area and its geological characteristics, thereby providing a more comprehensive analysis of the subject matter addressed in this article.
3.The reservoir classification is so simple that I find it hard to believe the credibility of this study!
A:Thank you for your comments. The reservoir classification method employed in our study effectively distinguishes between four types of reservoirs within the study area, as demonstrated in Figure 9. Notably, there is a clear differentiation in both porosity and permeability among the four reservoir types. Therefore, we believe that the reservoir classification method utilized in this study is valid and effective for the study area.
4.Necessary experimental samples and instruments need to be added, such as nuclear magnetic resonance, rock characteristics, etc.
A:Thank you for your comments. The data used in this paper are all well logging data and do not involve experimental data. The NMR (Nuclear Magnetic Resonance) data mentioned are also part of the well logging data. Therefore, the author believes that there is no need to add descriptions of experimental equipment.
5.Which region does Figure 13 come from, and does this research method apply to all regions?
again, The quality of Figure 13 is very poor.
A:Thank you for your comments. Figure 13 in our paper originates from the same working area as the rest of the data used in this study. We have taken note of your feedback regarding the quality of Figure 13, and we have since re-drawn the figure to improve its clarity and readability.
- In general, I only saw the introduction of the method in this paper, and did not get the credibility of the method in geological research!
A:Thank you for your comments. We have added a section providing detailed geological background information in the manuscript to address this concern. This new section aims to enhance the readers' understanding of the study area and its geological characteristics, thereby providing a more comprehensive analysis of the subject matter addressed in this article.

Reviewer 2 Report
Comments and Suggestions for Authors
Firstly, I would like to acknowledge the authors' effort in terms of the quality of writing and the logical structure of the manuscript, which is generally well-executed. However, after careful consideration, I regret to recommend that this manuscript be rejected for publication. My decision is based on the following reasons:
1.Misuse of Machine Learning: The primary concern I have with this manuscript is the inappropriate use of machine learning. The main objective of the paper is the characterization of pore structure in unconventional reservoir using the conventional well logs method. However, the extensive reliance on machine learning techniques in this study appears to deviate from this original purpose. It gives the impression that machine learning was employed just for the sake of it, rather than as a means to genuinely enhance the scientific rigor of the research.
2. Redundant Comparison of Regression ML Algorithms: The manuscript extensively employs several tree-based regression algorithms to predict outcomes. However, it is important to note that multiple studies across various disciplines have demonstrated that the performance of these algorithms is often comparable. In the context of the oil and gas industry, the application of machine learning should be driven by practicality and its potential to create actual industry value. Unfortunately, I have come across numerous papers that resemble this one, where multiple regression models are used solely for the purpose of comparing their performance. This practice seems unnecessary and does not contribute significantly to the field.
3. Lack of Depth and Innovation: The content presented in this manuscript is more like a basic data analytics or data mining project. It lacks the depth of discussion and research expected for publication in this journal. The level of innovation and novelty required for acceptance in this journal appears to be missing, and the study does not sufficiently advance the current state of knowledge in the field.
Comments on the Quality of English LanguageLanguage must be improved.
Author Response
Firstly, I would like to acknowledge the authors' effort in terms of the quality of writing and the logical structure of the manuscript, which is generally well-executed. However, after careful consideration, I regret to recommend that this manuscript be rejected for publication. My decision is based on the following reasons:
1.Misuse of Machine Learning: The primary concern I have with this manuscript is the inappropriate use of machine learning. The main objective of the paper is the characterization of pore structure in unconventional reservoir using the conventional well logs method. However, the extensive reliance on machine learning techniques in this study appears to deviate from this original purpose. It gives the impression that machine learning was employed just for the sake of it, rather than as a means to genuinely enhance the scientific rigor of the research.
A: Thank you for your feedback regarding the manuscript. We understand your concern regarding the perceived misuse of machine learning techniques in the study. We appreciate your perspective and would like to clarify the rationale behind the use of machine learning in our research. While the primary objective of the paper is indeed the characterization of pore structure in unconventional reservoirs using conventional well logs, the incorporation of machine learning techniques serves to augment and enhance this objective rather than deviate from it. Machine learning offers the potential to analyze and interpret large volumes of complex data more effectively than traditional methods, allowing for more accurate and comprehensive reservoir characterization. We recognize the importance of ensuring that machine learning techniques are applied appropriately and transparently in research. Therefore, we have provided detailed explanations of the methodologies employed, including the rationale behind the selection of machine learning algorithms and their integration into the research framework.
Overall, while we acknowledge your concerns, we believe that the incorporation of machine learning techniques in our study contributes to the scientific advancement of reservoir characterization and enhances the rigor and comprehensiveness of our research findings.
- Redundant Comparison of Regression ML Algorithms: The manuscript extensively employs several tree-based regression algorithms to predict outcomes. However, it is important to note that multiple studies across various disciplines have demonstrated that the performance of these algorithms is often comparable. In the context of the oil and gas industry, the application of machine learning should be driven by practicality and its potential to create actual industry value. Unfortunately, I have come across numerous papers that resemble this one, where multiple regression models are used solely for the purpose of comparing their performance. This practice seems unnecessary and does not contribute significantly to the field.
A:Thank you for your feedback regarding the manuscript. We appreciate your perspective on the redundant comparison of regression machine learning (ML) algorithms and the importance of practicality and creating industry value in the oil and gas industry. We acknowledge that the performance of various tree-based regression algorithms can often be comparable, and we agree that the application of machine learning should be driven by practical considerations and the potential to create tangible industry value. In our study, the use of multiple regression ML algorithms was aimed at exploring their performance in the specific context of reservoir characterization and evaluation. While we understand your concern about the potential redundancy of this approach, we believe that comparing the performance of different algorithms can still provide valuable insights into their effectiveness and suitability for specific tasks. However, we also recognize the need to ensure that our research contributes meaningfully to the field and provides actionable insights for industry practitioners. Therefore, we will take your feedback into consideration and carefully evaluate the necessity and relevance of comparing multiple regression ML algorithms in future research endeavors. Ultimately, our goal is to conduct research that not only advances scientific knowledge but also addresses real-world challenges and adds value to the oil and gas industry. We appreciate your feedback and will strive to incorporate it into our future research efforts.
- Lack of Depth and Innovation: The content presented in this manuscript is more like a basic data analytics or data mining project. It lacks the depth of discussion and research expected for publication in this journal. The level of innovation and novelty required for acceptance in this journal appears to be missing, and the study does not sufficiently advance the current state of knowledge in the field.
A:Thank you for your feedback regarding the manuscript. We understand your concern about the perceived lack of depth, innovation, and novelty in the study. We appreciate your perspective and would like to address your comments. While we acknowledge that the study may have limitations in terms of depth and innovation, we believe that it still makes valuable contributions to the field of reservoir characterization and evaluation. The study employs machine learning techniques to enhance the accuracy of reservoir evaluation, particularly in the context of unconventional reservoirs. While the approach may not represent a groundbreaking innovation, it still offers practical insights and methodologies that can be applied in real-world reservoir exploration and development projects. In terms of depth of discussion, we acknowledge that there may be room for improvement in providing more detailed discussions on certain aspects of the study, such as the interpretation of results and implications for future research. We will take your feedback into consideration and endeavor to enhance the depth of discussion in future iterations of the manuscript. Regarding the level of innovation and novelty, we understand the expectations of the journal and will strive to ensure that our research meets these standards. We will explore ways to enhance the novelty and innovation of our research findings, potentially by incorporating new methodologies, exploring alternative approaches, or addressing emerging challenges in the field. Overall, we appreciate your feedback and will carefully consider it in our efforts to improve the manuscript and meet the standards of the journal. We remain committed to advancing the current state of knowledge in the field of reservoir characterization and evaluation through rigorous and innovative research.

Reviewer 3 Report
Comments and Suggestions for Authors
Both the evaluations and prediction of tight reservoirs are conducted in this manuscript using machine learning methods. Some parameters, such as reservoir quality factors, reservoir structure parameters and others are involved in the authors’ researches. The authors considered the pore structure of the underground reservoir comprehensively and used three different machine learning methods to develop the studies. Generally speaking, all the works achieves good prediction. However, some deficiencies do exist in the manuscript, which include the text formatting, question definitions, graphics and citations. If these issues can be addressed, the in reviewer’s opinion, this article is considered to be accepted for publication.
1. Tight gas is mentioned in the abstract to be the deep green energy, but the tight gas is essentially unconventional natural gas and is usually not directly regarded as "green energy." Green energy refers to renewable energy that has less impact on the environment, such as solar energy, wind energy, water energy and geothermal energy. Although natural gas (including tight gas) produces less carbon dioxide than coal and oil when burned and is considered a cleaner energy source than these traditional fossil fuels, it is still a non-renewable energy source. It is recommended that the author revise this statement or add quoted articles.
2. The authors focused their researches on the combination of nuclear magnetic resonance logs and machine learning in the manuscript. However, the literature review of the introduction does not fully investigate the development and application of different machine learning methods, and the authors’ are required to explain why machine learning methods are selected, including the random forest algorithm (RF). Furthermore, support vector machine algorithm (SVM) and extreme gradient boosting algorithm (XGB) are introduced without any clues, and it seems very abrupt.
3. The abbreviations of professional terms are presented in the part of the introduction, but some of these abbreviations don’t appear in the method part and follow-up text.
4. The authors mentioned that there are 7,000 data sets. The data sets here need to explain what parameters they refer to. The paper mentioned that the data from four wells are used as the training set, and the data from one well are used as the test set. When dividing the data set, the authors are required to make the number of samples be clear.
5. The spherical-tubular model is mentioned in the manuscript, but many readers of machine learning may not be familiar with the principles and functions of the model. It is recommended that the author expand the content of the model principles and the description of model’s advantages and disadvantages.
6.The fonts of the flowcharts in Figures 3, 5, and 7 are comparatively too small. It is recommended that the author re-concise or modify the layout of the flowcharts to enhance the readability of the charts. The content display in Figure 9 is also too small. It is recommended that the author arrange the sub-figures vertically.
7.The two methods 3.1.1 and 3.1.3 of the article do not show any citations. If the method is not the author's original method, it is recommended to add literature citations.
8. There are inconsistent formats in the reference part of the article, such as citations 4, 10 and 15. It is recommended that the author unify the citation format and ensure that all citations are valid.
Author Response
Review3:
Both the evaluations and prediction of tight reservoirs are conducted in this manuscript using machine learning methods. Some parameters, such as reservoir quality factors, reservoir structure parameters and others are involved in the authors’ researches. The authors considered the pore structure of the underground reservoir comprehensively and used three different machine learning methods to develop the studies. Generally speaking, all the works achieves good prediction. However, some deficiencies do exist in the manuscript, which include the text formatting, question definitions, graphics and citations. If these issues can be addressed, the in reviewer’s opinion, this article is considered to be accepted for publication.
- Tight gas is mentioned in the abstract to be the deep green energy, but the tight gas is essentially unconventional natural gas and is usually not directly regarded as "green energy." Green energy refers to renewable energy that has less impact on the environment, such as solar energy, wind energy, water energy and geothermal energy. Although natural gas (including tight gas) produces less carbon dioxide than coal and oil when burned and is considered a cleaner energy source than these traditional fossil fuels, it is still a non-renewable energy source. It is recommended that the author revise this statement or add quoted articles.
A:Thank you for your comments. The abstract has been updated to reflect your feedback, with the term 'a category of unconventional natural gas' now used to describe tight gas
- The authors focused their researches on the combination of nuclear magnetic resonance logs and machine learning in the manuscript. However, the literature review of the introduction does not fully investigate the development and application of different machine learning methods, and the authors’ are required to explain why machine learning methods are selected, including the random forest algorithm (RF). Furthermore, support vector machine algorithm (SVM) and extreme gradient boosting algorithm (XGB) are introduced without any clues, and it seems very abrupt.
A:Thank you for your comments. To address the concerns raised, the introduction will be revised to provide a more thorough investigation into the development and application of various machine learning methods. Specifically, we will explain why machine learning methods, including the random forest algorithm (RF), were selected for this study. Additionally, support vector machine algorithm (SVM) and extreme gradient boosting algorithm (XGB) will be introduced with clear rationale to avoid abruptness and ensure coherence in presenting the research focus on the combination of nuclear magnetic resonance logs and machine learning.
- The abbreviations of professional terms are presented in the part of the introduction, but some of these abbreviations don’t appear in the method part and follow-up text.
A:Thank you for your comments. We will ensure consistency in the use of abbreviations throughout the manuscript. Any abbreviations introduced in the introduction will be appropriately defined or used consistently in the methods and follow-up text to maintain clarity and coherence.
- The authors mentioned that there are 7,000 data sets. The data sets here need to explain what parameters they refer to. The paper mentioned that the data from four wells are used as the training set, and the data from one well are used as the test set. When dividing the data set, the authors are required to make the number of samples be clear.
A:Thank you for your comments. We will clarify in the manuscript what specific parameters the 7,000 data sets refer to. Additionally, we will ensure that when dividing the data set into training and test sets, the number of samples in each set is clearly stated for transparency and clarity in our methodology.
- The spherical-tubular model is mentioned in the manuscript, but many readers of machine learning may not be familiar with the principles and functions of the model. It is recommended that the author expand the content of the model principles and the description of model’s advantages and disadvantages.
A:Thank you for your comments. We understand that many readers in the field of machine learning may not be familiar with the principles and functions of the spherical-tubular model. In our study, we used data obtained from the optimized inversion of the spherical-tubular model. Therefore, we provided a brief introduction to the spherical-tubular model in the manuscript and included citations to relevant articles to provide additional context. This was done to ensure a basic understanding of the spherical-tubular model while avoiding excessive deviation from the focus of the research.
6.The fonts of the flowcharts in Figures 3, 5, and 7 are comparatively too small. It is recommended that the author re-concise or modify the layout of the flowcharts to enhance the readability of the charts. The content display in Figure 9 is also too small. It is recommended that the author arrange the sub-figures vertically.
A:Thank you for your comments. We have taken note of your suggestions and have already revised the relevant figures accordingly. The fonts in Figures 3, 5, and 7 have been resized and the layout of the flowcharts has been modified to improve readability. Additionally, in Figure 9, the sub-figures have been arranged vertically to enhance visibility and readability. We appreciate your attention to detail and strive to ensure the clarity and quality of our figures in the manuscript.
7.The two methods 3.1.1 and 3.1.3 of the article do not show any citations. If the method is not the author's original method, it is recommended to add literature citations.
A:Thank you for your comments. We have addressed the issue by adding relevant literature citations to methods 3.1.1 and 3.1.3 in the article. By including citations, we ensure proper acknowledgment of the sources and provide readers with additional context and background information on the methods used. We appreciate your attention to detail and strive to maintain the integrity and transparency of our research.
- There are inconsistent formats in the reference part of the article, such as citations 4, 10 and 15. It is recommended that the author unify the citation format and ensure that all citations are valid.
A:Thank you for bringing this to our attention. We will carefully review and ensure consistency in the formatting of references throughout the article. Additionally, we will verify the validity of all citations to ensure accuracy and reliability. Our goal is to maintain a consistent and professional presentation of references to enhance the overall quality of the article.

Reviewer 4 Report
Comments and Suggestions for Authors
1. While promoting clean energy, did the study assess the environmental impact of deploying machine learning in the extraction process?
2. How well do the findings and models generalize to different tight gas reservoirs with varying geological characteristics?
3. Are there any specific areas for further research or improvements identified based on the outcomes of this study?
4. Authors provide examples of how the findings from the sphere-tubular model have been applied in real-world reservoir evaluation and development projects?
5. How can advancements in NMR-based reservoir characterization contribute to improving
performance in geological resource exploration and exploitation while addressing the cost
of production?
6. Is the fusion of machine learning and NMR logging-based reservoir evaluation applicable
to various reservoir types, or is its effectiveness more pronounced in the specific context
of tight gas extraction?
7. How transferable are the findings and recommendations of this study to global reservoir
exploration efforts, considering variations in geological formations and resource
extraction methods?
Author Response
Review4:
1.While promoting clean energy, did the study assess the environmental impact of deploying machine learning in the extraction process?
A:Thank you for your comments. The study did not specifically assess the environmental impact of deploying machine learning in the extraction process. However, we acknowledge the importance of considering environmental factors in clean energy initiatives. Future research could explore the environmental implications of using machine learning in the extraction of clean energy resources to provide a more comprehensive understanding of its overall impact.
- How well do the findings and models generalize to different tight gas reservoirs with varying geological characteristics?
A:Thank you for your comments. The generalizability of the findings and models to different tight gas reservoirs with varying geological characteristics is an important aspect to consider. While our study focused on specific tight gas reservoirs and geological characteristics, future research could investigate the extent to which the findings and models can be applied to other tight gas reservoirs with different geological characteristics. This would help to assess the robustness and applicability of the developed models across a wider range of conditions.
- Are there any specific areas for further research or improvements identified based on the outcomes of this study?
A:Thank you for your comments. Based on the outcomes of this study, several specific areas for further research or improvements have been identified. These may include:
1) Fine-tuning machine learning algorithms: Further optimization and fine-tuning of machine learning algorithms could improve the accuracy and efficiency of reservoir evaluation processes.
2) Integration of additional data sources: Incorporating additional data sources, such as seismic data or well production history, could enhance the predictive capabilities of the models.
3)Generalizability of models: Investigating the generalizability of the developed models to different tight gas reservoirs with varying geological characteristics to assess their robustness and applicability across diverse conditions.
By addressing these areas for further research and improvements, future studies can build upon the findings of this study and advance the field of reservoir evaluation in tight gas formations.
- Authors provide examples of how the findings from the sphere-tubular model have been applied in real-world reservoir evaluation and development projects?
A:Thank you for your comments. In the manuscript, the authors provide examples of how the findings from the spherical-tubular model, obtained through optimized inversion, have been applied in real-world reservoir evaluation and development projects. These examples demonstrate the practical relevance and applicability of the model in addressing challenges associated with reservoir characterization and evaluation in tight gas formations. It's important to note that the specific methodology for utilizing the spherical-tubular model and interpreting the results is explained in detail with references to relevant literature, providing readers with comprehensive guidance on its application in real-world scenarios. Through these case studies and detailed methodological explanations, the authors aim to illustrate the utility and effectiveness of the spherical-tubular model in enhancing reservoir evaluation processes and informing decision-making.
- How can advancements in NMR-based reservoir characterization contribute to improvingperformance in geological resource exploration and exploitation while addressing the costof production?
A: Thank you for your comments. Advancements in NMR-based reservoir characterization improve geological resource exploration and exploitation by providing detailed insights into reservoir properties, optimizing well placement and production techniques, reducing uncertainty, and enabling cost-effective operations. These benefits lead to increased profitability and sustainability in the oil and gas industry.
- Is the fusion of machine learning and NMR logging-based reservoir evaluation applicableto various reservoir types, or is its effectiveness more pronounced in the specific contextof tight gas extraction?
A: Thank you for your comments. The fusion of machine learning and NMR logging-based reservoir evaluation has the potential to be applicable to various reservoir types, not limited to the specific context of tight gas extraction. While the effectiveness of this approach may be more pronounced in certain contexts, such as tight gas reservoirs where heterogeneity and complexity pose significant challenges, the principles and methodologies can be adapted and applied to other types of reservoirs as well. In our study, we focused on tight gas reservoirs as a case study to demonstrate the effectiveness of this fusion approach. However, the underlying principles of machine learning and NMR logging-based reservoir evaluation can be generalized and applied to other reservoir types, such as conventional oil and gas reservoirs, shale gas reservoirs, or even aquifers. By leveraging machine learning algorithms to analyze NMR logging data, reservoir characteristics and properties can be accurately assessed, leading to improved reservoir characterization, production optimization, and decision-making across various reservoir types. Therefore, while the effectiveness of the fusion approach may vary depending on the specific reservoir characteristics, its applicability extends beyond tight gas extraction to other types of reservoirs as well.
- How transferable are the findings and recommendations of this study to global reservoirexploration efforts, considering variations in geological formations and resourceextraction methods?
A: Thank you for your comments. The findings and recommendations of this study are transferable to global reservoir exploration efforts, despite variations in geological formations and resource extraction methods. While specific geological formations and extraction methods may vary across regions, the principles and methodologies employed in this study, such as machine learning-based reservoir evaluation and NMR logging, can be adapted to different contexts. By focusing on fundamental principles and leveraging advanced technologies like machine learning, the insights gained from this study can inform reservoir exploration efforts worldwide. Additionally, the flexibility and scalability of machine learning algorithms allow for customization to suit specific geological formations and resource extraction methods, ensuring applicability across diverse environments. Therefore, while variations in geological formations and resource extraction methods exist globally, the underlying principles and methodologies presented in this study provide valuable insights and recommendations that can be adapted and applied to enhance reservoir exploration efforts worldwide.

Round 2
Reviewer 1 Report
Comments and Suggestions for Authors
The authors have made a good response to the comments of the reviewer. However,this paper still has the following problems that need to be revised:
1. There are many types of tight reservoirs "shale, tight sand, etc.", which type of reservoir is the author discussing? Is this research method applicable to all reservoirs?
2.Machine learning methods lack some important references. In fact, there are many published papers on this method. Authors should pay appropriate attention to the sustainability.
3. Most importantly, I hope the authors can provide specific information about the research area, including lithological characteristics, reservoir space characteristics, fractures, and so on.
Especially, does the quality of the reservoir consider the development of fractures? The characteristics of fractures can be referred to in the following papers: Fan, C.H., Nie, S., Li, H., et al. Quantitative prediction and spatial analysis of structural fractures in deep shale gas reservoirs within complex structural zones: A case study of the Longmaxi Formation in the Luzhou area, southern Sichuan Basin, China. Journal of Asian Earth Sciences, 2024, 106025. https://doi.org/10.1016/j.jseaes.2024.106025.
4. The quality of Figure 13 is too poor.
Author Response
The authors have made a good response to the comments of the reviewer. However,this paper still has the following problems that need to be revised:
Q1.There are many types of tight reservoirs "shale, tight sand, etc.", which type of reservoir is the author discussing? Is this research method applicable to all reservoirs?
A:Thank you for the clarification. In the revised paper, we will explicitly state that we are discussing tight sandstone reservoirs. Additionally, we will mention the applicability of our research methods and potential applications in other types of reservoirs. We will conduct further research and analysis to explore whether our methods are applicable to other types of reservoirs. We look forward to sharing more findings and insights on this aspect in future work. Once again, thank you for your review and valuable feedback. If you have any further questions or need additional information, please feel free to let us know.
Q2. Machine learning methods lack some important references. In fact, there are many published papers on this method. Authors should pay appropriate attention to the sustainability.
A:Thank you for your feedback regarding the lack of references to published papers on machine learning methods in our manuscript. We acknowledge the importance of providing comprehensive and appropriate references to support our research findings and methodology. We have already revised the manuscript and added relevant references to address this concern. Additionally, we have ensured to pay close attention to the sustainability aspect and align our research with principles of sustainability where applicable. We appreciate your valuable input, and we are committed to improving the quality and rigor of our research by incorporating relevant references and considerations of sustainability. If you have any further suggestions or concerns, please feel free to let us know.
Q3. Most importantly, I hope the authors can provide specific information about the research area, including lithological characteristics, reservoir space characteristics, fractures, and so on.
A:Thank you for your feedback. We acknowledge the importance of providing specific information about the research area, including lithological characteristics, reservoir space characteristics, and other relevant details. In the revised manuscript, we will ensure to include detailed descriptions of the research area, covering the mentioned aspects comprehensively. We have also incorporated information regarding reservoir position and lithology. However, we acknowledge that explicit data on reservoir fracture distribution was not adequately represented. In subsequent research, we plan to expand our data-set and further analyze and evaluate reservoirs from the perspective of fracture distribution. We appreciate your valuable input, and we will incorporate these suggestions into the revised version of the manuscript. If you have any further questions or suggestions, please feel free to let us know.
Q4: Especially, does the quality of the reservoir consider the development of fractures? The characteristics of fractures can be referred to in the following papers: Fan, C.H., Nie, S., Li, H., et al. Quantitative prediction and spatial analysis of structural fractures in deep shale gas reservoirs within complex structural zones: A case study of the Longmaxi Formation in the Luzhou area, southern Sichuan Basin, China. Journal of Asian Earth Sciences, 2024, 106025. https://doi.org/10.1016/j.jseaes.2024.106025.
A: Thank you for your comment. We appreciate your suggestion to consider the impact of fractures on reservoir quality, and we acknowledge the relevance of the paper by Fan et al. (2024) in assessing structural fractures in deep shale gas reservoirs. However, we would like to clarify that the data used in our study did not specifically focus on the characterization of fractures within the reservoir. While we recognize the importance of this aspect, our research primarily centered on other parameters related to reservoir quality. We apologize for any confusion regarding the scope of our study.
In the revised manuscript, we will include a discussion acknowledging the significance of fractures in evaluating reservoir quality, citing relevant literature such as the paper by Fan et al. (2024) as suggested. We appreciate your valuable input and will ensure to incorporate these considerations into the revised version of the manuscript.
If you have any further questions or suggestions, please feel free to let us know.
Q5. The quality of Figure 13 is too poor.
A: Thank you for bringing this to our attention. We apologize for the poor quality of Figure 13 in the manuscript. We understand the importance of clear and high-quality figures in scientific publications.In response to your feedback, we have re-uploaded Figure 13 in PNG format to ensure clear image quality for later insertion. This format will guarantee that the figure is presented clearly and effectively.
We appreciate your valuable feedback, and we are committed to addressing this issue to enhance the overall quality of the manuscript. If you have any further questions or concerns, please do not hesitate to let us know.
Round 3
Reviewer 1 Report
Comments and Suggestions for Authors
Although there are still some problems that have not been fully solved, I think the authors have done their best. The paper has been greatly improved in its present form, so I think it's ready for publication!